# Enzyme-assisted high throughput sequencing of an expanded genetic alphabet at single base resolution

Bang Wang[1,2], Kevin M. Bradley[3], Myong-Jung Kim[3], Roberto Laos[1], Cen Chen [1], Dietlind L. Gerloff[1], Luran Manfio[1], Zunyi Yang [1,3] ✉ & Steven A. Benner [1,3] ✉

With just four building blocks, low sequence information density, few functional groups, poor control over folding, and difficulties in forming compact folds, natural DNA and RNA have been disappointing platforms from which to evolve receptors, ligands, and catalysts. Accordingly, synthetic biology has created "artificially expanded genetic information systems" (AEGIS) to add nucleotides, functionality, and information density. With the expected improvements seen in AegisBodies and AegisZymes, the task for synthetic biologists shifts to developing for expanded DNA the same analytical tools available to natural DNA. Here we report one of these, an enzyme-assisted sequencing of expanded genetic alphabet (ESEGA) method to sequence six-letter AEGIS DNA. We show how ESEGA analyses this DNA at single base resolution, and applies it to optimized conditions for six-nucleotide PCR, assessing the fidelity of various DNA polymerases, and extending this to AEGIS components with functional groups. This supports the renewed exploitation of expanded DNA alphabets in biotechnology.

A standard challenge in biotechnology arises from our inability to design molecules from first principles to meet the performance needed for biotechnological applications. Proteins, in principle, could deliver "performance on demand"; natural protein evolution does this for a spectacularly broad range of functions. However, computationally intensive protein design,[1] as well as protein-targeted laboratory evolution[2] require enormous amounts of trial and error, as well as knowledge of thousands of pre-solved structures. Further, outside of privileged scaffolds (antibodies are exemplary), the enormous sequence space of proteins is dominated by molecules that do not fold or, worse, precipitate from water. Folding and dissolution in water are nearly universal requirements for biotechnological value.

Nucleic acids (DNA and RNA) have better-defined folding rules. Further, they remain soluble throughout their sequence spaces due to their repeating backbone charges[3], and enjoy direct evolvability without the intermediacy of complex ribosome-based translation. RNA catalysts may have supported life during an episode of its early

evolution, the "RNA World"[4]. Accordingly, pioneers like Larry Gold, Jack Szostak, Gerald Joyce, and others suggested that nucleic acids might be platforms for laboratory in vitro evolution (LIVE) to create functional biopolymers[5].

Unfortunately, three decades of effort with LIVE on natural scaffolds have been often disappointing[6,7]. This disappointment has been attributed to the low information density of standard DNA/RNA (which hinders defined folding), their lack of functional groups needed for efficient binding and catalysis, and the intrinsic difficulty of getting compact core folds from their polyanionic backbone.

These limitations might be mitigated in DNA analogs that exploit alternative hydrogen bonding patterns to give "artificially expanded genetic information systems" (AEGIS, Fig. 1)[8]. For example, adding non-standard nucleobases adds alternative base–base interactions that dramatically expand the number of compact folds available to evolving AEGIS oligonucleotides. These include isoG pentaplexes[9] (with the one letter code B), fat and skinny duplexes[10], and the recently

[1]Foundation for Applied Molecular Evolution, Alachua, FL, USA. [2]Department of Chemistry, University of Florida, Gainesville, FL, USA. [3]Firebird Biomolecular Sciences, LLC, Alachua, FL, USA. ✉e-mail: zyang@ffame.org; sbenner@ffame.org

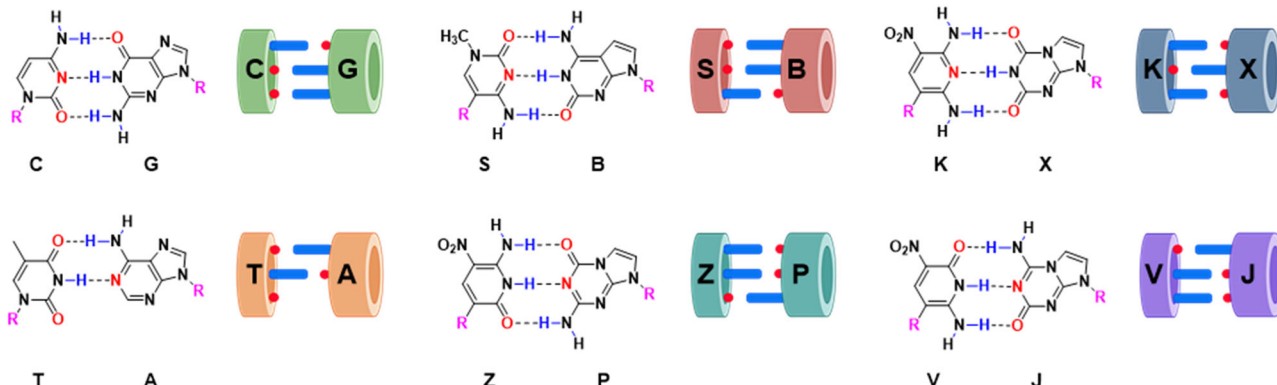

**Fig. 1 | Chemical structures of standard and non-standard nucleobases.** By rearranging hydrogen bonding donor and acceptor groups on base pairs in a Watson–Crick geometry, the number of independently replicable informational units in DNA/RNA can be increased from 4 to 12, increasing the information density, functionality, and density of binders and catalysts in libraries of oligonucleotides built from an artificially expanded genetic information system (AEGIS).

reported fZ-motif[11]. This last fold exploits the low p$K_a$ of Z to give "skinny" deprotonated Z⁻:Z pairs in a novel parallel double helix.

Consistent with this, AEGIS-LIVE is proving to be a useful alternative to phage display and computationally intensive design for proteins, and as an alternative for systematic evolution of ligands by exponential enrichment for standard nucleic acids. Evolved AEGIS-bodies, antibody analogs, inactivate toxins[12], bind cancer cell surface proteins[13,14], and deliver drugs selectively to targeted malignant cells[15]. AEGIS libraries from six-nucleotide AEGIS DNA (G, A, C, T, Z, P, Fig. 1) are at least 100,000 times richer than standard GACT libraries as reservoirs for GACTZP AEGISzyme ribonucleases, analogs of protein ribonucleases[16]. This is due to the ability of Z to act as a general acid-base catalyst. No comparable activity is seen with any standard nucleobase.

The challenge to support AEGIS-LIVE now is to develop methods that efficiently sequence six-nucleotide (GACTZP) AEGIS DNA. Since manufacturers of "next generation" sequencing instruments have not been persuaded to directly sequence non-standard AEGIS components of DNA, controlled transliteration of AEGIS DNA to standard DNA has been at the core of these methods.

Previously in these laboratories, a sequencing method was developed that integrates both an "easy" and a "difficult" transliteration[17]. The "easy transliteration" occurs when Z:P is converted to C:G through pairing between deprotonated Z and G. The Z⁻:G pair has a Watson–Crick geometry, allowing it to evade many proof-reading mechanisms. This makes it "easy".

In contrast, a "difficult" transliteration requires T or C to pair opposite P. Neither is easy at standard PCR pHs, and therefore is not clean. This second transliteration means Z:P pairs are transliterated to a mixture of T:A and C:G pairs. The ratio in this mixture is very sensitive to conditions, making the bioinformatics analysis challenging, and preventing the analysis of very complex mixtures.

In collaboration with Andrew Laszlo's teams, we have recently published preliminary data suggesting the possibility of using nanopores to sequence expanded genetic alphabets[18]. Similar approaches have been studied for hydrophobic unnatural nucleotides[19]. Nevertheless, these approaches remain in their infancy.

Sequencing approaches of other unnatural nucleotide sets have suffered from similar challenges. For example, dye terminator Sanger sequencing[20] with low throughput, the similar transliteration strategies[21–23] were applied to hydrophobic pairs with NGS[24]. Li and coworkers recently reported a clever transliteration strategy for Romesberg's TPT3-NaM pair[25]. However, pairing between hydrophobic and hydrogen-bonding nucleobases, required for transliteration, need not always support quantitative sequencing results.

Thus, to fully realize the potential of AEGIS, we need reliable, efficient, quantitative, and user-friendly methods to sequence GACTZP DNA. We report here such a method: enzyme-assisted sequencing of the expanded genetic alphabet (ESEGA). Rather than using transliteration during PCR, we enzymatically transform a starting mix to convert all cytidines to uridines using a member of the "Apolipoprotein B mRNA editing catalytic polypeptide-like" (AID/APOBEC[26]) deaminase family[27]. APOBEC converts standard cytidine (C) in an oligonucleotide to uridine (U), a deterministic transliteration that occurs in high yield. Separately, we exploit the relatively low p$K_a$ (≈7.8) of AEGIS Z, which in its deprotonated form mismatches with G. This allows clean transliteration of Z:P pairs to C:G pairs during PCR. Finally, since no standard nucleobase effectively mismatches with P, we develop a workflow that incorporates dZTP into transliterative PCR to make the only necessary mismatch in the workflow between deprotonated Z and G.

Together, these approaches allow us to exploit the power of next-generation sequencing (NGS)[28] instruments. These deliver millions of reads from single samples for four-letter DNA. The final part of the workflow is bioinformatics. After deamination and five-nucleotide PCR conversion, a comparison of the results of deep sequencing of AEGIS PCR products, in parallel, with antisense and sense DNA, allows bioinformatics to infer the sequences of AEGIS-containing molecules in the starting mixture, even complex mixtures that arise from AEGIS-LIVE.

To demonstrate its utility, we use ESEGA to (1) define optimal conditions to perform 6-triphosphate PCR conditions (such as buffer pH and dPTP concentrations). (2) Evaluate the 6-triphosphate PCR fidelity with various commercial and house-engineered DNA polymerases. (3) Extend that evaluation to functionalized AEGIS components, in particular, those with alkynyl and aromatic hydrophobic functional groups, which are sparsely introduced into AEGIS libraries because of the higher information density of a six-letter GACTZP DNA alphabet.

## Results

### Developing ESEGA

To develop ESEGA sequencing, two single-stranded DNA sequences were synthesized to serve as test beds. These were accompanied by control sequences made from standard nucleotides ("Nat"), and a Z-modified sequence, where C was replaced by Z in the natural sequence (Table 1). The "Nat" sequence contains two restriction sites that are recognized by two restriction endonucleases, AluI (AGCT) and PspOMI (GGGCCC). The ZZ trial sequence contains Zs placed strategically so that if they are transliterated to C, the AluI and PspOMI sites

**Table 1 | Standard and AEGIS DNA sequences used in this study**

| Name | Sequence (5'–3') |
|---|---|
| Nat | TAAGATGAGAGTTGAGGAGAGTTATCCAAGCTATAGGGCCCTTCAGTATAGTAGTGTAAGTAGATAGTGGA |
| ZZ | TAAGATGAGAGTTGAGGAGAGTTATCCAAGZTATAGGGCZZTTCAGTATAGTAGTGTAAGTAGATAGTGGA |
| ZP-1 | TAAGATGAGAGTTGAGGAGAGTTACGTGZACGCPTPGTCAZCACAGTATAGTAGTGTAAGTAGATAGTGGA |
| ZP-2 | TAAGATGAGAGTTGAGGAGAGTTATCAPCGTAGCAZPCTTPTZATGTATAGTAGTGTAAGTAGATAGTGGA |
| C-Ran | TAAGATGAGAGTTGAGGAGAGTTATNNNCNNNGTATAGTAGTGTAAGTAGATAGTGGA |
| Z-Ran | TAAGATGAGAGTTGAGGAGAGTTATNNNZNNNGTATAGTAGTGTAAGTAGATAGTGGA |
| P-Ran | TAAGATGAGAGTTGAGGAGAGTTATNNNPNNNGTATAGTAGTGTAAGTAGATAGTGGA |

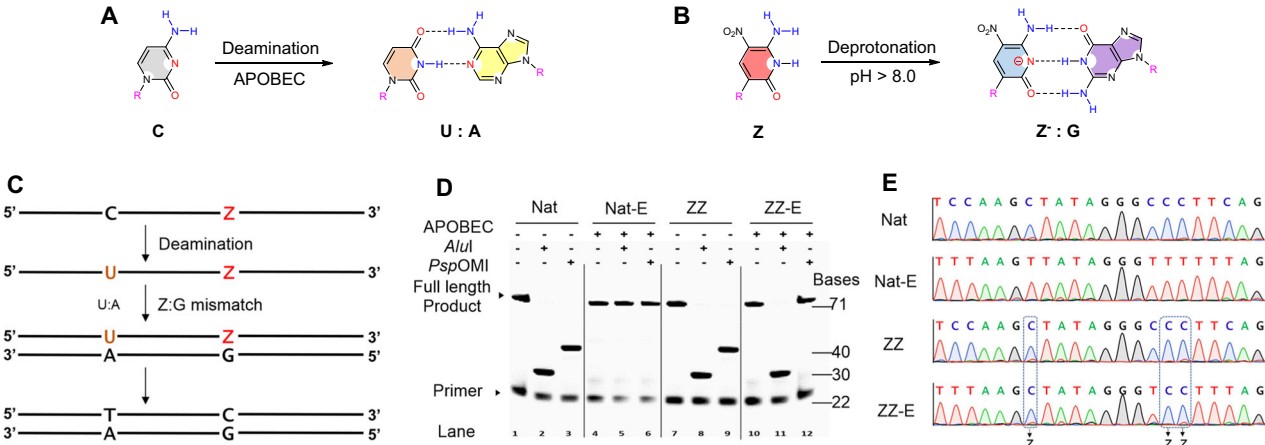

**Fig. 2 | Sequencing five-letter AEGIS (ATCGZ) DNA by deamination and transliteration. A** Cytidine (C) is transliterated by cytidine deaminase to form uridine (U), which pairs with A in PCR. **B** AEGIS base Z becomes Z⁻ at pH (8.9) by deprotonation; Z⁻ pairs with G during PCR, inducing a Z to C transliteration. **C** Schematic workflow shows the C to T and Z to C conversions after deamination and PCR amplification. **D** Denaturing PAGE-urea analysis of restriction digestion of PCR products by TakaRa Taq DNA polymerase at pH 8.9 from DNA templates (Nat and ZZ) without deamination or with deamination (Nat-E and ZZ-E). Forward primer was labeled by FAM at 5'. The uncropped gel was provided in the Source data file. The experiment was repeated three times independently, yielding similar results. **E** Sanger sequencing demonstrates the precise transliteration of C to U (then T during the PCR) and Z to C.

are re-generated (Supplementary Fig. 1). This transliteration can be detected by strategic restriction digestion (Fig. 2D).

To develop and metric ESEGA sequencing, samples of both Nat and ZZ sequences were treated with cytidine deaminase; controls were not treated. Then, treated and untreated sequences were PCR amplified (pH 8.9) in mixtures containing only four standard dNTPs (no dZTP, no dPTP "4-triphosphate PCR",). These conditions force template dZ to misdirect the incorporation of dGTP (Fig. 2B).

As the pH of PCR buffer may affect PCR efficiency, a series of pH values of PCR buffer (from 7.4 to 9.5, measured at room temperature) were evaluated by quantitative PCR (qPCR); $C_q$ values were used as metrics. Both Nat and ZZ templates were well amplified between pH 8.0 and 9.3 (Supplementary Fig. 4). As the preferred PCR conditions to facilitate Z → C transliteration, pH 8.9 was chosen. PCR products were then digested by AluI or PspOMI. The PCR amplicons from the natural template without deaminase treatment gave one well-identified length digestion band in denatured urea-PAGE analysis in lanes 2 and 3 (Fig. 2D), as expected from faithful amplification of the two sites in the synthetic standard DNA.

In contrast, PCR amplicons from the standard template that had been previously treated by deaminase ("Nat-E") resisted restriction digestion (Fig. 2D, lanes 5 and 6). This showed that the deaminase completely converted the Cs to Us in the restriction sites (Fig. 2A); these appear as T in the PCR amplicons (Fig. 2C).

When the ZZ template was amplified in PCR with just four standard triphosphates, amplicons were also well digested by endonucleases (Fig. 2D, lanes 8 and 9). This showed that Z is converted to C during the PCR amplification. With the ZZ template treated with cytidine deaminase ("ZZ-E") and then amplified by PCR, amplicons were digested by AluI (lane 11). This indicated that: (i) Z is not affected by cytidine deaminase; (ii) an isolated Z can be successfully transliterated to C.

However, the ZZ-E amplicons resisted the digestion by PspOMI (Fig. 2D, lane 12). This suggested that the PspOMI restriction site was changed from GGCZZ to GGTCC by deamination of the C to U and transliterating ZZ to CC. This also showed that C→U deamination by APOBEC was not affected by a neighboring ZZ.

To confirm this by Sanger sequencing, the length of the sequencing DNA was extended by tagged PCR (from 71 bp to 323 bp, Supplementary Fig. 2). The sequencing results (Fig. 2E) agree well with the restriction digestion. The sequence of the "Nat" amplicons matched with original design. The sequences of the Nat-E amplicons showed that all the Cs were deaminated. In the ZZ template, all of the Zs were transliterated to C by PCR (at pH 8.9). For the ZZ-E sample, the sequences showed that the original Cs were completely transliterated to Ts (Fig. 2E). However, they also show three Cs signals arising from the positions originally holding Z, either isolated Z or consecutive ZZ.

We then investigated how DNA sequences built from six nucleotides "letters" (A, C, T, G, Z, P) were amplified under different PCR conditions. Two other test AEGIS DNA molecules were designed to contain both Z and P (ZP-1 and ZP-2, Table 1) and synthesized. For DNA sequences containing P to work when only standard (A, T, C, G) triphosphates are present, P is forced to mismatch with either C or T during the initial PCR cycles. However, this encounters problem with conversion of P, since all mismatches available in this amplification are incompatible with the Watson–Crick geometry (Fig. 3A, left arrow).

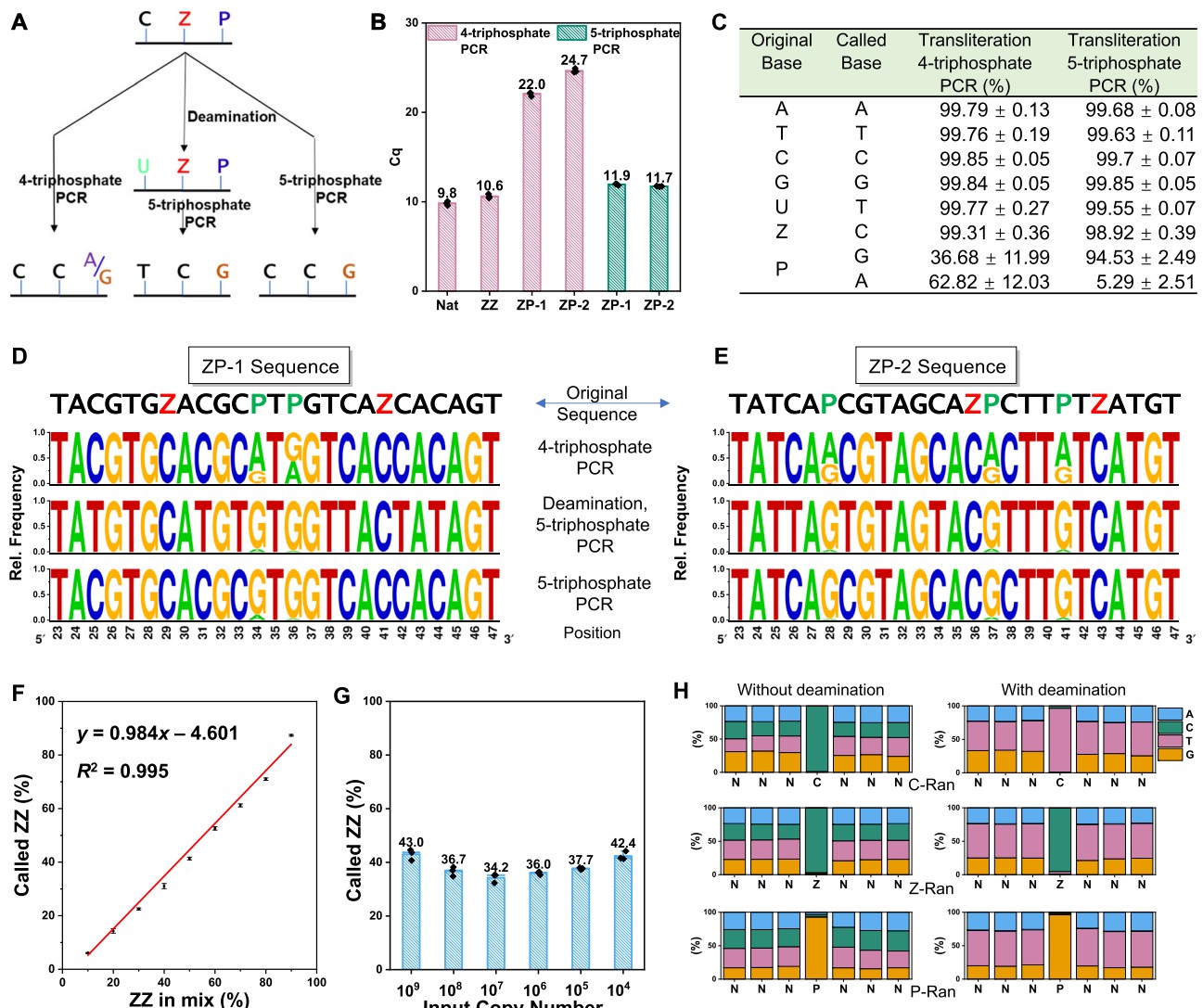

**Fig. 3 | Enzyme-assisted sequencing of expanded genetic alphabet (ESEGA) of single strand six-letter DNA (ATCGZP).** Complete NGS read summaries are in the Supplemental material. **A** Schematic diagram showing the transliteration of AEGIS (ATCGZP) DNA using 4-triphosphate or 5-triphosphate PCR at pH 8.9, without or with deamination. **B** Quantitative PCR (qPCR) to evaluate the transliteration efficiency of various DNA templates in 4-triphosphate and 5-triphosphate PCR. Error bars represent the standard deviation of three independent experiments ($n = 3$; mean ± SD). **C** NGS data reveals the transliteration (%) of original bases [A, T, C, (U), G, Z, and P] in Nat, ZZ, ZP-1, and ZP-2 sequences under different PCR conditions (4-triphosphate or 5-triphosphate PCR at pH 8.9, with or without deamination). Standard deviations represent the multiple bases in the three sequences. **D, E** Sequence logos of six-letter DNA (ZP-1 and ZP-2) under different transliteration

and PCR conditions. **F** Robustness and specificity of the ESEGA for ZZ and Nat sequences at different ratios (10%, 20%, 30%, 40%, 50%, 60%, 70%, 80%, and 90%). The error bars represent the standard deviation of three independent DNA base measurements ($n = 3$; mean ± SD). **G** Sensitivity of ESEGA for ZZ (45%) and Nat (55%) sequences in a series of 10-fold dilutions. The error bars represent the standard deviation of three independent DNA base measurements ($n = 3$; mean ± SD). **H** Robustness of ESEGA with libraries NNNCNNN (C-Ran, top), NNNZNNN (Z-Ran, middle), and NNNPNNN (P-Ran, bottom) sequences, without deamination (left panel) or with deamination (right panel), then, followed by 5-triphosphate PCR and NGS. These data show that ESEGA has no strong context dependency. Note: all statistical source data are available in the "Source Data" file.

Isolated Zs and Ps in ZP-1 were paired and read in primer extension experiments (Supplementary Fig. 3C). qPCR analysis showed that the $C_q$ values of ZP-1 ($C_q = 22.0$) were higher than those of the Nat sequence (9.8) and ZZ sequences ($C_q = 10.6$), indicating the problematic nature of P:C and P:T mismatches (Fig. 3B). Further, when Z and P were adjacent (ZP-2), primer extension was completely inhibited (Supplementary Fig. 3D), and the $C_q$ values of ZP-2 in the 4-triphosphate PCR were even higher ($C_q = 24.7$) (Fig. 3B). This suggest that P mismatching to T or C is more problematic when Z is adjacent.

This poor mismatching was mitigated by adding dZTP to the four standard dNTPs (5-triphosphate PCR) (Fig. 3A, right arrow). This allows P to match with Z in the first PCR round. The Z in its deprotonated form

then directs the mismatched incorporation of G, leading to cleaner conversion. Thus, the ZP-2 sequence works very well in 5-triphosphate primer extension (Supplementary Fig. 3E, F). Both ZP-1 and ZP-2 sequences show high efficiency in 5-triphosphate PCR, the $C_q$ value (11.9 and 11.7) are close to Nat and ZZ template in 4-triphosphate PCR (Fig. 3B).

To obtain quantitative metrics for the fidelity of converting sequences built from six-letter (A, C, T, G, Z, P) DNA to sequencable nucleotides under different PCR conditions, the performance of ZP-1 and ZP-2 conversion was compared with these pre-treatments:

(1)   Direct amplification with 4-triphosphate PCR. This was expected to proceed with low efficiency with some ambiguous transliteration (Fig. 3A, left arrow).

(2) Treatment with deaminase, followed by amplification with 5-triphosphate PCR. This was expected to deliver high-efficiency PCR, with the clean conversion of Z:P to C:G, with all of the C:G in the original templates replaced by U:A, and then T:A pairs (Fig. 3A, central arrow).

(3) Direct amplification with 5-triphosphate PCR. This was expected to deliver high-efficiency PCR, with clean transliteration of Z:P to C:G, and with all of the C:G in the original templates remaining as C:G pairs (Fig. 3A, right arrow).

These amplicons were sent to Sanger sequencing (Supplementary Figs. 9 and 10) and NextGen sequencing. The analysis of NGS data revealed very faithful transliteration (>99%) of the original bases [ATC(U)GZ] in the Nat, ZZ, ZP-1, and ZP-2 sequences to their corresponding bases (A, T, C, and G) in 5-triphosphate or 4-triphosphate PCR. However, in the case of P transliteration, a mixture of A (~63%) and G (~37%) was observed in 4-triphosphate PCR, with relatively larger fluctuations. Fixing this problem, P was transliterated almost exclusively to G (~94.5%) in 5-triphosphate PCR (Fig. 3C). To visualize the transliteration of the base at a given position, we converted the table information into a sequence logo (Fig. 3D, E), which illustrates the transliteration of ZP-1 and ZP-2 templates under for 4-triphosphate PCR (top), 5-triphosphate PCR following deamination (middle), and 5-triphosphate PCR without deamination (bottom).

To test the robustness of ESEGA, ZZ, and Nat templates were mixed with total concentrations of 100 nM, but with AEGIS-containing and standard oligonucleotides in various ratios (10%, 20%, 30%, 40%, 50%, 60%, 70%, 80%, and 90%). Subsequently, each sample underwent a single-stranded ESEGA with NGS.

The called populations of ZZ amplicons displayed a modest reduction of approximately 4.5% relative to the prepared ZZ percentage across all the templates (Fig. 3F).

This observation might be attributed to the slightly lower efficiency of $Z^-$ mismatch with G in the first round of PCR compared with the standard DNA base pair. This is supported by the higher $C_q$ value recorded in qPCR for the ZZ template (10.6) in contrast to the Nat template (9.8) (Fig. 3B).

To evaluate the sensitivity of ESEGA, ZZ (45%) and Nat templates (55%) were blended and subsequently diluted serially to give six different concentrations and inputs ranging from $10^9$ to $10^4$ copies in sequencing. Each sample was subjected to a single-stranded AEGIS-DNA sequencing. Sequencing was possible even at the highest dilution (Fig. 3G).

So far, the cytidine deamination and Z/P conversion were demonstrated with defined six-letter DNA sequences. We were concerned that the local sequence context might influence the outcome delivered by ESEGA. To determine whether neighboring nucleotides may affect C to U, Z to C, and P to G transliteration, three sequences were synthesized where a single C, Z, or P was placed in the middle of six random nucleotides (C-Ran, Z-Ran, and P-Ran, Table 1). The ESEGA workflow was applied. Low sequence bias was seen in the deamination results (Fig. 3H, top), consistent with the literature[29]. Further, no overall sequence context bias was observed in Z and P transliteration (Fig. 3H, middle and bottom).

Double-stranded DNA is a common outcome of AEGIS six-nucleotide PCR. Thus, we developed an ESEGA workflow for double-stranded DNA as well. First, the double-stranded DNA was denatured and the strands were separated. The two single-strands were separately treated with deaminase followed by 5-triphosphate PCR. The two amplicon pools were separately sequenced with a barcode. Bioinformatics then matched the sequences to the strands that were originally paired. Then, the matches were analyzed to infer the original sequences of the paired strands (Fig. 4A). A:T and T:A pairs delivered A:T and T:A pairs in the duplexes matched by bioinformatics analysis, unchanged by the processes in the workflow. Thus, sites that hold A:T and T:A pairs in the surviving bioinformatics pairs were inferred to have been A:T and T:A pairs in the original duplex.

Likewise, Z:P and P:Z in the original duplex gave C:G and G:C in the bioinformatics-paired sites. In both cases, they arise by transliteration involving deprotonate Z:G mismatches. Thus, sites that hold G:C and C:G pairs in the surviving bioinformatics pairs were inferred to have been Z:P and P:Z pairs in the original duplex.

If the original duplexes have C:G or G:C pairs, then bioinformatics analysis gives a third outcome due to deamination. From amplicons arising from the strand that contained C, deamination gives amplicon duplexes with T:A pairs. From the complementary strand that contained G, the amplicon duplexes hold G:C pairs at the homologous site. Thus, bioinformatics assigns C:G in the original duplex when A:T appears in the amplicons derived from the "sense" DNA chain, if C:G also appears in amplicons derived from the anti-sense DNA chain.

## Determining six-nucleotide PCR conditions that optimally retain Z:P pairs

To illustrate how ESEGA might be used in a practical setting, we first showed how ESEGA applied to double-stranded amplicons might be used to evaluate various concentrations of dPTP and various values of pH values the impact on the fidelity of GACTZP PCR. Here, the metric for fidelity was the percent retention of Z:P pairs when the ZP-1 sequence was used as a template.

FAM-labeled forward and Cy5-labeled reverse primers were used to amplify ZP-1 in PCR with a complete set of six triphosphates, with a primer: template ratio of 50,000:1 (~16 nominal doublings). TaKaRa Taq HS polymerase and 6-triphosphate dNTP (dATP (0.1 mM), dCTP (0.2 mM), dGTP (0.1 mM), dTTP (0.1 mM), dZTP (0.1 mM), and dPTP) were used in the amplification. Six different dPTP concentrations (0.05 mM, 0.1 mM, 0.2 mM, 0.3 mM, 0.4 mM, and 0.5 mM) and two pH levels (8.0 and 8.9) were used, under the hypothesis that Z:P pairs would be better retained at higher concentrations of dPTP and better retained at lower pH (Supplementary Fig. 11).

Following the separation of the PCR duplex amplicons by PAGE-urea, ESEGA was used to compare the data from the sense and anti-sense strands to quantify the retention of the Z and P nucleotides after 25 rounds of PCR. Consistent with the hypothesis, Z:P pairs were better retained during PCR at pH 8.0 (Fig. 4C) than at pH 8.9 (Fig. 4B) at each concentration of dPTP. This was attributed to greater deprotonation of Z at the higher pH, leading to more deprotonated $Z^-$:G mismatches. Greater retention of the Z:P pairs was also observed with increasing dPTP concentration. This was consistent with the hypothesis that dPTP competes with dGTP as a partner for template dZ.

Thus, ESEGA supported an application for screening PCR conditions to identify parameters that best retained Z:P pairs. Under these conditions, ~90% of the Z:P pairs were retained at pH 8.0 with 0.5 mM dPTP after 16 nominal doublings, with a nominal per cycle fidelity of 99.34%.

## Identifying polymerases that optimally retain Z:P pairs

Using these optimal conditions (pH 8.0, 0.5 mM dPTP), we then amplify the ZP-1 template with a set of polymerases, including TaKaRa Taq HS, KOD exo⁻, KlenTaq, Phire hot start II, Phusion TM, Go Taq, One Taq, and an in-house-engineered 6 M Taq variant. All of these gave amplification and interpretable sequencing data. The amount of each polymerase was adjusted to ensure similar amplification efficiency. Other polymerases examined (LongAmp Taq, Q5 High-Fidelity, Sulfolobus, Vent exo⁻, and HiFi KAPA) produced inconsistent results or no amplification at all.

ESEGA was used to analyze amplicons from 25 cycles of 6-triphosphate PCR using these eight polymerases and the ZP-1 template. The retention rates of Z:P pairs were visualized using a sequence logo (Fig. 4D).

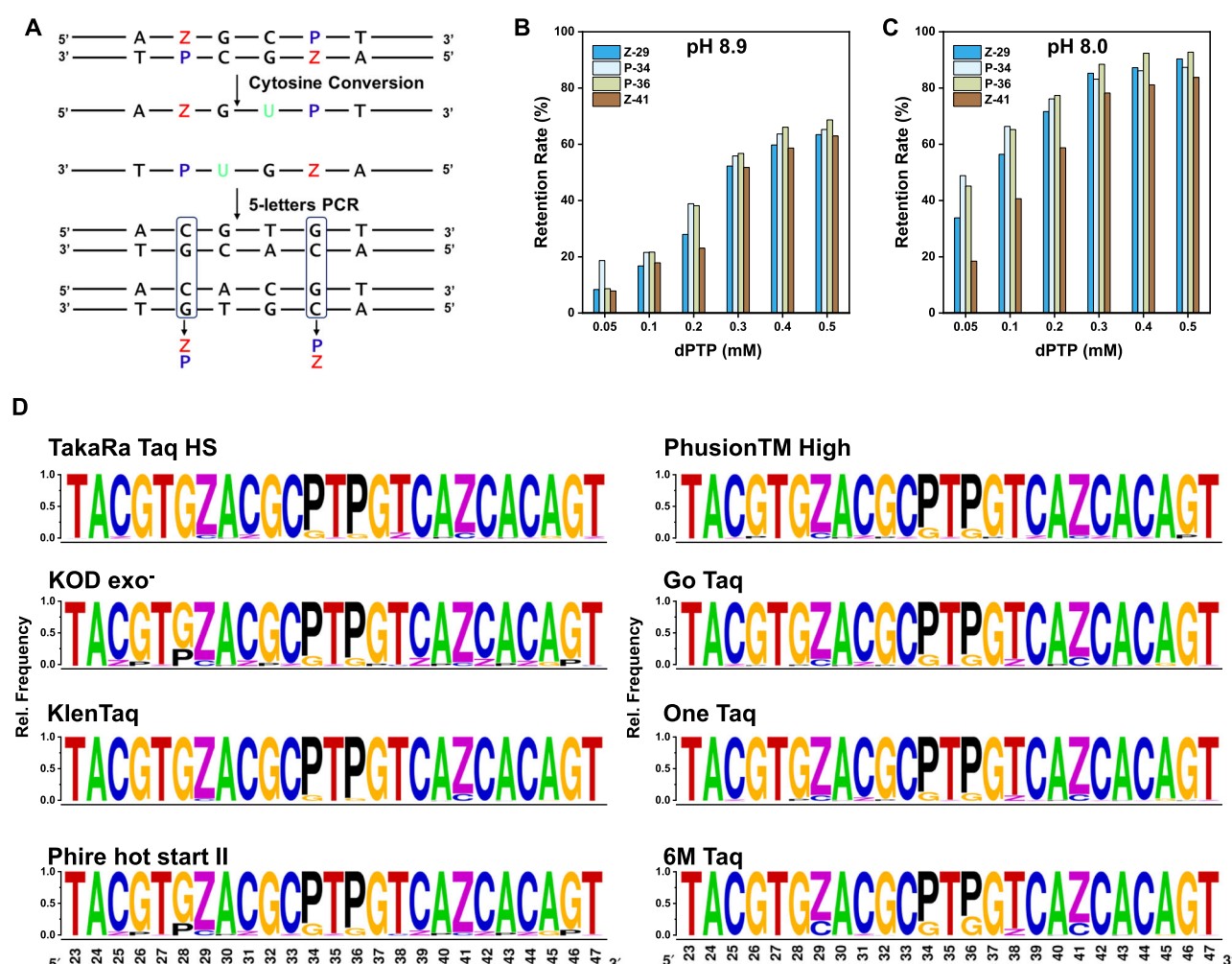

**Fig. 4 | Sequencing double-stranded six-letter DNA and assessing the fidelity of different DNA polymerases in 6-triphosphate PCR. A** Schematic of ESEGA workflow for double-stranded six-letter (ATCGZP) DNA. **B**, **C** Retention rates of Z/P at each position in the ZP-1 template plotted after 6-triphosphate PCR with varying concentrations of dPTP (0.05–0.5 mM) at pH 8.9 or pH 8.0 conditions. **D** ESEGA evaluation of the fidelity of various polymerases in 6-triphosphate PCR. Complete NGS read summaries are in the Supplemental material. Note: all statistical source data are available in the "Source data" file.

Our findings revealed that Z:P pairs were retained best by the KlenTaq polymerase under these conditions, retaining 90–95% of the Z:P pairs after 16 nominal doublings; this approximates the uncertainty in the ESEGA analysis itself. However, the KlenTaq polymerase gave less efficient amplification. Thus, TaKaRa Taq HS was identified as a preferred enzyme under a metric that combined fidelity, efficiency, and robustness.

Additionally, we observed that KOD exo⁻ polymerase exhibited relatively good fidelity in the retention of Z and P, but added Z and P to the amplicons at positions that originated as C and G. dZTP can potentially pair with G, which may lead to Z:P pairs replacing C:G pairs in six-letter PCR.

The 6 M Taq polymerase, which was developed in-house to encourage processivity, performed less well. A comprehensive description of the 6 M Taq evolution process can be found in the Supplementary materials. Overall, ESEGA provides a robust and reliable framework for the selection and development of high-fidelity polymerases in the context of 6-triphosphate PCR applications.

**Assessing the fidelity of functionalized dPTP in 6-triphosphate PCR**

As noted in the introduction, one of the advantages of AEGIS-LIVE over LIVE with standard nucleotides is the increased information density of expanded genetic alphabets, and the consequent ability to sparsely introduce functional groups in AEGIS-LIVE that standard DNA/RNA lacks. This allows AEGIS-LIVE to compete with protein evolution (e.g., phage display) and protein computational design (e.g., ROSETTA) by increasing the diversity of functional groups towards that of proteins, and increasing the number of compact folds, without the troublesome features of proteins, in particular, their propensity to precipitate.

Figure 5B, C shows two variants of AEGIS P that carry functional groups, specifically, ethynyl and phenylbutynyl groups. The first can support "click chemistry"; proteins have no analogous capability. The second is able to support hydrophobic interactions (compared with phenylalanine in proteins).

We used ESEGA to evaluate the performance of polymerases challenged to amplify AEGIS DNA containing these functionalized P variants. The ZZ sequence was chosen as a template, with functionalized dPTP used in the triphosphate mix instead of normal dPTP. The amplification was done as before with six-nucleotide PCR. The sense DNA chain was separated from resulting PCR products by PAGE-urea, and the sequences of the amplicons was evaluated by ESEGA, using both Sanger sequencing (Fig. 5D) and NGS (Fig. 5E). The six-nucleotide PCR was also monitored by qPCR (Eva green) when the fluor-labeled primers were replaced by unlabeled primers (Supplementary Fig. 12B).

Here, the efficiency of amplification by TaKaRa of oligonucleotides containing alkynyl P was close to that of those with unfunctionalized P. Amplification efficiency was modestly lower with

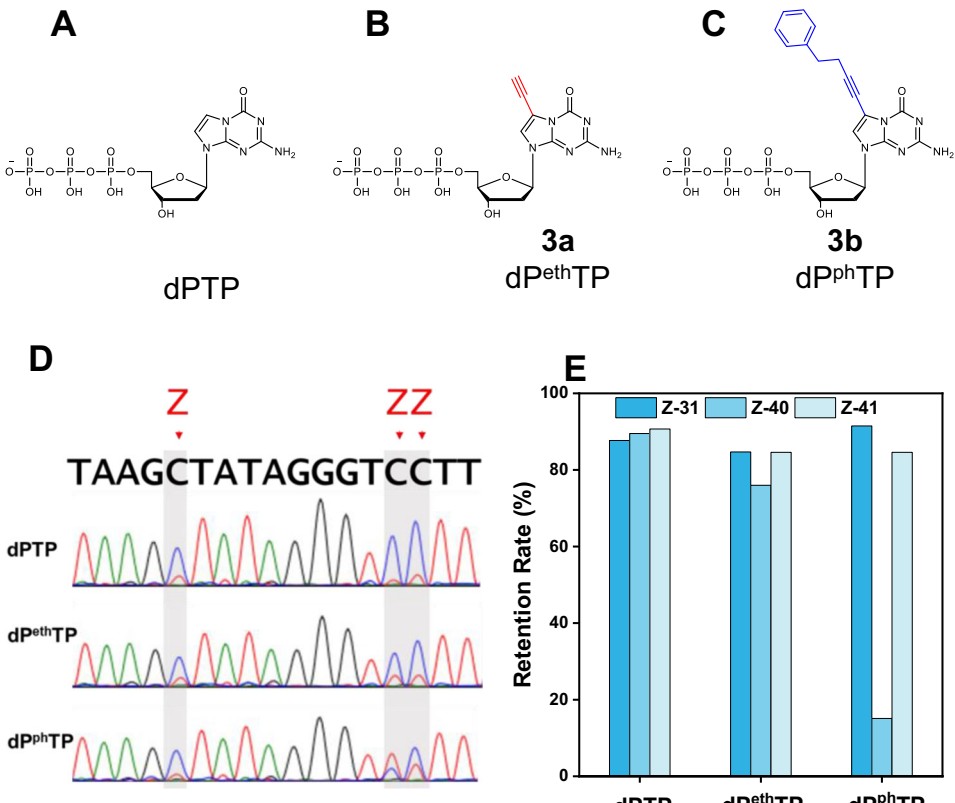

**Fig. 5 | Chemical structures and retention rate of functionalized-dPTP derivatives in 6-triphosphate PCR assessed by ESEGA.** Chemical structures of **A** classical dPTP; **B** dP^eth^TP, 7-(4Ethynyl)-dP triphosphate; and **C** dP^ph^TP. 7-(4-Phenyl-1-butynyl)-dP triphosphate. **D** Sanger sequencing PCR amplicons from ZZ template with 6-triphosphate (dNTPs, dZTP, and dPTP, or dP^eth^TP, or dP^ph^TP). **E** ESEGA and NGS evaluate the retentions of dP, dP^eth^, and dP^ph^ in PCR amplicons after 6-triphosphate PCR by statistics of Z retentions. Note: all statistical source data are available in the "Source data" file.

phenylalkynyl P (Supplementary Fig. 12B). The fidelity of replication was comparable with alkynyl P by ESEGA (Fig. 5E). However, substantial loss of phenylalkynyl P was observed by ESEGA, especially at position 40 of the template. Here, the two adjacent Zs drive the insertion of two tagged P's. It is well known that unmodified polymerases such as TaKaRa do not easily synthesize DNA with two consecutive tagged nucleotides[30]. Thus, this result was not unexpected.

The Z-Ran templates (-NNNZNNN-) were also used to evaluate the fidelity of six-nucleotide PCR involving various dPTP analogs. The procedures are similar to those described above. After six-nucleotide PCR, the FAM-labeled oligonucleotide products were separated by PAGE-urea and then submitted for ESEGA sequencing. The results show under standard dPTP conditions, the PCR fidelity, measured as Z retention, was 94.0%. However, when ethynyl-dPTP was substituted for standard dPTP, PCR fidelity decreased to 83.5%. With the introduction of phenylbutynyl-dPTP, the fidelity was further reduced to 81.2% (Supplementary Fig. 13).

## Discussion

Nearly all life forms on Earth share the same informational biopolymers using A–T and C–G base pairs. Even the known exceptions, cyanoviruses that use diaminopurine instead of adenine as a partner for T, do not expand the number of independently replicable building blocks[31–33].

However, standard nucleic acids lack the functional group diversity, the informational density, and the folding capability needed to give effective receptors, ligands, and catalysts. These factors account for the inability of LIVE with standard DNA and RNA to compete effectively with antibody and laboratory protein evolution, even though proteins lacking a privileged scaffold are plagued by precipitation issues.

Artificially expanded genetic information systems (AEGIS, Fig. 1) are not designed for any specific purpose, but rather to be richer reservoirs of functionality in a directly evolvable system. AEGIS has more building blocks, a greater diversity of functional groups, higher information density, better control over folding, and the ability to form compact folds via base–base interactions. Even with the limited sequencing tools previously available, AEGIS-LIVE has evolved molecules that neutralize toxins, cleave specific RNA molecules, bind to specific cells, and deliver drugs to cancer tissues[34,35]. Alternative systems where pairing does not exploit inter-base hydrogen bonding have been explored by Kool[36], Hirao[22], and Romesberg[37]. All have been shown to perform, at various levels of efficiency, in replication, transcription, translation, and semi-synthetic organisms[8,38,39]. More exotically, AEGIS is helping us to seek alien life in the cosmos[40], which may not have had the same pre-history as life on Earth, and thus may have different genetic biopolymers.

Therefore, AEGIS has the potential for broad biotechnological applications, should its evolution under selective pressures chosen by experimentalists become routine[41]. ESEGA offers a key element needed to make AEGIS-LIVE routine.

ESEGA represents a transformative use of the capabilities afforded by NSG, which has transformed the analysis of standard DNA sequences. By manipulating the pH level to alter the topological structure of nucleic bases, Z is deprotonated to form Z⁻, Z⁻ equates to C (Fig. 1). This allows Z⁻ to pair with G quite well, leading to high-fidelity transliteration (99%). Additionally, ESEGA employs 5-triphosphate transliteration PCR to ensure high-fidelity transfer of P to G (-95%), this ensures a clean outcome.

Standard bioinformatics allows hundreds of thousands of reads from NGS to be analyzed in a library context. The cleanliness of this

workflow supports high sequence diversity in those libraries, where each individual sequence is present in the mix as only a few dozen exemplars.

In addition to showing this robust workflow, we show three applications where ESEGA supports the development of synthetic biology using expanded DNA alphabets. These are, of course, not the only three that can be conceived. Thus, ESEGA-like workflows hold the potential to develop other expanded genetic alphabets.

## Methods

### Synthesis of model DNA oligonucleotides

All DNA sequences used in this project are listed in the Supplementary Table 1. Standard DNA templates and primers without AEGIS were purchased from Integrated DNA Technologies (Coralville IA). DNA templates containing dZ and/or dP base were provided by Firebird Biomolecular Sciences LLC (Alachua FL), synthesized on a Mermade 12 DNA synthesizer following standard phosphoramidite chemistry protocols. dZ or dP phosphoramidites are also from Firebird. Oligonucleotides were purified by ion exchange HPLC[42] (Dionex DNAPac™ PA-100 column, 9 × 250 mm) and desalted over SepPak C18-cartridges. DNA concentrations were quantified by Nanodrop One (Thermo Scientific). Detailed information on DNA sequences, as well as comprehensive details about commercial reagents and company names, can be found in the Supplementary section.

### The general protocol of sequencing by transliteration of AEGIS GACTZP DNA

(i)  The workflow for single-stranded AEGIS-DNA sequencing involves denaturing the DNA, enzymatically deaminating cytidine to give uracil, and PCR amplification at pH 8.9. The resulting PCR products are then sequenced using Sanger or NSG. A parallel experiment without cytidine deamination is performed to serve as a control. The resulting DNA sequence information from data arising with and without deamination are compared computationally to match deaminated and non-deaminated strands. From this comparison, the positions of Z and P in the parent sequences are inferred.

(ii) The workflow for double-stranded AEGIS-DNA sequencing involves the initial separation of the double-stranded DNA into two single-stranded DNA, followed by DNA denaturation and enzymatic deamination of cytidine to uracil. During subsequent five-triphosphate PCR amplification at pH 8.9. Uracil is transliterated to thymine, Z is transliterated to C, and P is transliterated to G. The resulting PCR products are then subjected to NSG. The DNA sequence information is deduced by comparing sense and antisense strands.

### Enzymatic conversion of cytidine to uracil

The cytidine deamination procedure was conducted following guidelines provided in the Enzymatic Methyl-seq kit instruction manual (New England Biolabs). Briefly, DNA denaturation was initiated by mixing 20 μl of the DNA sample (at a concentration of 1 μM) with 4 μl of formamide, followed by incubation at 85 °C for 10 min. Subsequently, the sample was rapidly cooled on ice and subjected to cytidine deamination through the addition of 100 μl of a pre-prepared master mix containing the APOBEC enzyme. The mixture was then incubated at 37 °C for 3 h. Following this step, the resulting DNA sample was purified using either bead-based or column-based methods, or directly heated (without purification) at 90 °C for 60 s. Finally, the DNA sample was diluted and prepared for subsequent PCR amplification.

### PCR amplification with standard or AEGIS triphosphates

The routine PCR system comprised DNA polymerase, PCR reaction buffer, dNTP, primers, template, and nuclease-free water. Three types of PCR were employed in this study, each differing in the type and concentration of dNTP used: regular 4-triphosphate PCR. dATP (0.2 mM), dCTP (0.2 mM), dGTP (0.2 mM), and dTTP (0.2 mM). Five-triphosphate PCR. dATP (0.2 mM), dCTP (0.2 mM), dGTP (0.2 mM), dTTP (0.2 mM), and dZTP (0.05 mM). Six-triphosphate PCR. dATP (0.1 mM), dCTP (0.2 mM), dGTP (0.1 mM), dTTP (0.1 mM), dZTP (0.1 mM), and dPTP (0.5 mM). qPCR reactions were conducted using a CFX96 real-time PCR system (BioRad).

### Restriction digestion and PAGE-urea analysis of PCR products

The Nat or ZZ templates were subjected to either deamination or left untreated, followed by a 4-triphoshate PCR (pH 8.9) amplification with FAM-labeled forward primer and unlabeled reverse primer. The PCR products were treated with either 1 μl of *Alu*I (10 U/μl) or 1 μl of *Psp*OMI (20 U/μl) restriction enzyme in a 10 μl reaction mixture to digest the amplified DNA templates. The digestion mixtures were then incubated at 37 °C for 4 h or overnight, followed by the addition of 2 μl of denaturing buffer (saturated urea solution) and heating at 95 °C for 30 s. The resulting products were then analyzed by 15% PAGE-urea-TBE, and the gel was visualized using a Typhoon imaging system (Amersham Biosciences) with the Cy2 channel.

### Extending the length of amplicons for Sanger sequencing

DNA templates were subjected to either deamination or left untreated, followed by performing length extension. Briefly, a total reaction volume of 50 μl was prepared, consisting of 5 μl of A1-200 (500 nM, 200 bp), 5 μl of B1-100 (500 nM, 100 bp), 1 μl DNA template, 25 μl of 2× PCR master mix (TaKaRa Taq), and 14 μl of nuclease-free water. The mixture was subjected to an initial extension and pre-amplification step under thermocycling conditions (95 °C for 1 min, 95 °C for 15 s, 60 °C for 10 s, and 72 °C for 30 s, for 10 cycles). The resulting pre-amplified products (5 μl) were then transferred to a new tube containing a mixture of A1 primer (0.5 μM), B1 primer (0.5 μM), and PCR master mix (TaKaRa Taq pH 8.9). The mixture was subjected to a standard PCR amplification step under thermocycling conditions (95 °C for 1 min, 95 °C for 15 s, 58 °C for 10 s, and 72 °C for 30 s, for 12 cycles). The resulting PCR products were analyzed by 20% native PAGE in TBE buffer and visualized by Sybr-gold staining and Typhoon imaging system (Amersham Biosciences) at the Cy2 channel (Supplementary Fig. 2). The amplified products were then purified by 2% agarose gel using the GeneJET gel extraction kit (ThermoFisher) and submitted for Sanger sequencing by Genewiz.

### $^{32}$P γ-ATP radiolabel DNA primer

A reaction mixture (total volume of 20 μl in a 200 μl tube) comprised 12 μl of nuclease-free water, 2 μl of primer (10 μM), 2 μl of T4 polynucleotide kinase reaction buffer (10×), 1 μl of T4 polynucleotide kinase (10 U/μl), and 3 μl of [γ-$^{32}$P] ATP. The mixture was gently mixed, followed by incubation at 37 °C for 30 min. The labeled DNA was purified by a standard purification kit (QIAquick Nucleotide Removal Kit).

### Primer extension

A reaction mixture (total volume 20 μl) was prepared in a 100 μl PCR tube containing unlabeled reverse primer (50 nM), $^{32}$P γ-ATP labeled reverse primer (10 nM), template DNA (300 nM), and PCR master mix (containing TaKaRa Taq polymerase, reaction buffer pH 8.9, and 0.2 mM d(A, T, C, G)TP). For the 5-triphosphate primer extension, an additional dZTP (0.1 mM or 0.05 mM) was added to the mixture. The resulting mixture was subjected to primer extension under these conditions: 95 °C for 30 s, 51 °C for 30 s, and 72 °C for varying times (10 s, 30 s, 60 s, 120 s, and 180 s). The extension was stopped by adding a quenching mix buffer containing 4 μl EDTA (2 mM) and urea (8 M), followed by heating at 95 °C for 60 s using a thermocycler. The resulting products were analyzed using a 15% TBE-urea PAGE (7 M

urea), and the gel was exposed to a phosphor screen for 2 h before being visualized by autoradiography (Supplementary Fig. 3).

## PCR under varying pH conditions

PCR buffers with varying pH values were prepared in-house using a 10× recipe consisting of 100 mM Tris-HCl, 500 mM KCl, and 15 mM MgCl$_2$. The pH of each buffer was adjusted by adding 1 M HCl, and the resulting pH values (7.4, 7.8, 8.0, 8.5, 9.0, 9.3, and 9.5) were monitored using an electronic pH meter. The prepared buffer solutions were then filtered through a 0.22 μM nylon syringe filter. For qPCR, a reaction mixture with a total volume of 50 μl was prepared in each tube, consisting of 0.25 μl of TaKaRa Taq HS (5 U/μl), 5 μl of PCR buffer (10×), 4 μl of dNTP mixture (A, T, C, G, 2.5 mM each), 5 μl of template (50 pM), 5 μl of R-Primer (5 μM), 5 μl of F-Primer (5 μM), and 1 μl of EvaGreen dye (20×), with the remaining volume consisting of nuclease-free water. The resulting mixture was subjected to standard PCR amplification under thermocycling conditions (95 °C for 1 min for pre-heating, 95 °C for 15 s, 56 °C for 10 s, and 72 °C for 30 s, for 30 cycles) monitored by SYBR/FAM channel. Each condition of the qPCR experiment was carried out in triplicate. It is worth noting that EvaGreen® dye (manufacturer's labeling as 20×) was used as 50× in the qPCR reaction (Supplementary Fig. 4).

## qPCR analysis ten-fold diluted template

Each reaction (total volume of 50 μl) comprised of 0.25 μl TaKaRa Taq HS (5 U/μl), 5 μl PCR buffer (10×), 4 μl dNTP mixture (either 4-triphosphate or 5-triphosphate), 5 μl template (ten-fold diluted from copy number of $10^9$–$10^3$), 2.5 μl R-Primer (5 μM), 2.5 μl F-Primer (5 μM), and 1 μl of EvaGreen dye (20×), with the remaining volume consisting of nuclease-free water. The resulting mixture was subjected to standard PCR amplification under thermocycling conditions (95 °C for 1 min for pre-heating, 95 °C for 15 s, 56 °C for 10 s, and 72 °C for 30 s, for 30 cycles) monitored by SYBR/FAM channel. Each condition of the qPCR experiment was carried out in triplicate. The standard curve was constructed using the mean $C_q$ values obtained (Supplementary Fig. 5).

## NSG of AEGIS DNA

The single-strand AEGIS DNA templates were subjected to either deamination or left untreated, followed by performing length extension and PCR amplification with barcode and NGS adapter. In the process of length extension, a reaction volume of 50 μl was prepared, consisting of 5 μl of A1-100 (500 nM, 100 bp), 5 μl of B1-100 (500 nM, 100 bp), 1 μl of AEGIS DNA template, 25 μl of 2× PCR master mix (TaKaRa Taq, pH 8.9), and 14 μl of nuclease-free water. The mixture underwent an initial extension and pre-amplification step under thermocycling conditions (95 °C for 1 min, 95 °C for 15 s, 60 °C for 10 s, and 72 °C for 30 s, for 10 cycles). After length extension, the template was prepared for NGS using the PCR-mediated addition of Illumina adapter sequences. Prior to sample processing, forward and reverse primers were designed to incorporate a unique eight-base barcode sequence, aimed at reducing the risk of contamination during the NGS sequencing process. The primers were structured as follows: 5'-Illumina adapter-8 base barcode-extend DNA primer-3'. The exact sequence information of the primers can be found in Supplementary Table 2. The length-extended products (5 μl) were then transferred to a new tube containing a mixture of barcoded primers (0.5 μM), and PCR master mix (TaKaRa Taq), and underwent standard PCR amplification under thermocycling conditions (95 °C for 1 min, 95 °C for 15 s, 58 °C for 10 s, and 72 °C for 30 s, for 12–20 cycles). The PCR process was monitored by Evergreen fluorescence. The PCR products were then analyzed by 15% PAGE-urea/TBE and purified with 2% agarose gel using the QIA gel extraction kit. Finally, the purified products were submitted for NGS analysis by GENEWIZ (Amplicon-EZ).

## Bioinformatics analysis of NGS data

The NGS sequence analysis was performed using a combination of traditional software packages and custom in-house scripts. Briefly, sequence reads were merged using USEARCH v11.0.667[43]. These reads were then filtered to those that contained the correct internal barcodes. Using Bowtie2 v2.3.2[44], the reads were aligned to the target of interest, with parameters that allowed for multiple mismatches to capture any level of base changes due to deamination. Pileup files were then created using SAMtools v1.15.1[45] to detail the base variability at each position. These files were then used in a custom script that detailed the percentage of bases present at each position, which were compared against the expected location of AEGIS bases.

## Assessing the robustness of ESEGA through correlation tests

The ZZ sequence and Nat sequence templates were mixed at concentrations of 100 nM in various percentage ratios (10%, 20%, 30%, 40%, 50%, 60%, 70%, 80%, and 90%). Subsequently, each sample underwent a single-stranded AEGIS-DNA sequencing workflow consisting of denaturation, deamination, and four-triphosphate PCR (pH 8.9) length extension, followed by barcoding PCR amplification. The resulting barcoded PCR products were purified using the QIA gel extraction kit using a 2% agarose gel, and then subjected to NSG by GENEWIZ.

## Evaluating the sensitivity of ESEGA-sequencing

The ZZ sequence sample (45%) and Nat sequence sample (55%) were mixed at a concentration of 100 nM and further diluted ten-fold into varying concentrations. Each sample with input copy numbers of $10^9$, $10^8$, $10^7$, $10^6$, $10^5$, and $10^4$ underwent a single-stranded AEGIS-DNA sequencing workflow, which included denaturation, deamination, and four-triphosphate PCR (pH 8.9) length extension, followed by barcoding PCR amplification. The resulting barcoded PCR products were purified using the QIA gel extraction kit with 2% agarose gel and subsequently submitted for NGS servers at GENEWIZ. This sensitivity analysis aimed to assess the performance and accuracy of the ESEGA-sequencing method across various template input copy numbers, highlighting the potential for precise and reliable ESEGA-sequencing.

## Assessing the sequence dependence of ESEGA-sequencing

DNA templates, including C-Ran (-NNNCNNN-), Z-Ran (-NNNZNNN-), and P-Ran (-NNNPNNN-), were subjected to either deamination or left untreated, followed by single-stranded AEGIS DNA sequencing procedures. These procedures involved length extension and PCR amplification using barcoded NGS adapters. The agarose gel-purified PCR products were submitted for NGS analysis by GENEWIZ. During the bioinformatics analysis process, a custom script was employed to extract the 3, 2, and 1 flanking bases for all sequences at the location of possible transliteration. Subsequently, these extracted bases were categorized as either having the expected base or not.

## Double-strand DNA sequencing for various concentrations of dPTP at pH 8.0 or pH 8.9

A reaction volume of 50 μl was prepared, comprising a FAM-labeled forward primer (500 nM) and a Cy5-labeled reverse primer (500 nM) which separated by an internal C3 spacer and Poly-T, template ZP-1 (10 pM), 6-triphosphate dNTP (dATP (0.1 mM), dCTP (0.2 mM), dGTP (0.1 mM), dTTP (0.1 mM), dZTP (0.1 mM), and dPTP (0.05–0.5 mM)), TaKaRa Taq HS polymerase (1.25 U), and pH 8.0 or 8.9 PCR buffer. The solution was subjected to the 6-triphosphate PCR application under thermocycling conditions (95 °C for 1 min, 90 °C for 15 s, 54 °C for 10 s, and 72 °C for 60 s, for 24 cycles). The resulting PCR duplexes products were separated by 15% PAGE-urea, and the gel was scanned using a Typhoon imaging system (Amersham Biosciences) at Cy2 and Cy5 channels. The corresponding gel bands (sense strand and anti-sense

strand DNA) were excised and collected into 50 ml PP centrifuge tubes with 30 ml of 0.2 M TEAA buffer and incubated overnight. The solution was then desalted using fresh SepPak-C18 cartridges. The resulting DNA solution was lyophilized, dissolved in a 50 μl water solution, and then subjected to single-strand AEGIS-DNA sequencing respectively. The FAM-labeled DNA chain was extended by A1-100 and B1-100 DNA using 5-triphosphate PCR conditions after deamination, while the Cy5-labeled DNA chain was extended by C1-100 and D1-100 DNA using 5-triphosphate PCR conditions after deamination. The product then underwent barcoding PCR amplification and agarose gel purification, and was submitted to NGS. The AEGIS DNA sequencing information was inferred by comparing these two DNA chains. The Z/P percentage was determined using the formula $Y = 0.984X - 4.601$ and NGS statistical data (Supplementary Fig. 11).

### ESEGA evaluates the fidelity of 6-triphosphate PCR using different DNA polymerases

A reaction mixture (50 μl) was prepared consisting of a FAM-labeled forward primer (500 nM) and a Cy5-labeled reverse primer (500 nM), template ZP-1 (10 pM), a 6-triphosphate dNTP mixture (dATP: 0.1 mM; dCTP: 0.2 mM; dGTP: 0.1 mM; dTTP: 0.1 mM; dZTP: 0.1 mM; and dPTP: 0.5 mM), pH 8.0 PCR buffer and polymerase. By adjusting the amount of polymerase to equilibrium the PCR efficiency of diverse polymerase. TaKaRa Taq HS (2 U), KOD exo- (0.8 U), KlenTaq (9 U), Phire hot start II (2 U), Phusion TM (1.5 U), Go Taq (2 U), One Taq (1.5 U), 6 M Taq (4 U). The solution was subjected to the 6-triphosphate PCR application. A total of 13 polymerases were utilized, of which eight exhibited successful performance, as confirmed by PAGE gel analysis. The resulting PCR duplexes products were separated by 15% PAGE-urea and subjected to AEGIS-DNA sequencing. The produce refers to double-strand DNA sequencing (Supplementary Fig. 11).

### Expression and purification of 6 M Taq DNA polymerase variant

The 6 M (six positions mutation) Taq DNA polymerase lacks the first 280 amino acids, and it is fused with the DNA binding protein of Sulfolobus solfataricus on the N-terminus. This enzyme was evaluated based on formal reports[46] to prevent Z:P mismatches. The protein sequence is listed in Supplementary information.

Briefly, the expression of 6 M Taq DNA polymerase ("6 M") was achieved by transforming a plasmid, pJExpress414, containing the gene encoding 6 M, into *Escherichia coli* C43(DE3) using NheI and NcoI restriction sites. A single colony was cultured in LB medium (50 ml) supplemented with 0.5 mM IPTG and incubated at 37 °C with shacking overnight. On the subsequent day, the cellular pellet was harvested via refrigerated centrifugation at 5000× g for 10 min and stored at −20 °C. The pellet was then resuspended in a pH 8.0 solution containing 50 mM glucose, 50 mM Tris, and 1 mM ethylenediaminetetraacetic acid (EDTA), subjected to heat-shock at 75 °C for 10 min, and combined with BugBuster® Master Mix (Novagen) in a 2:3 ratio for 20 min at room temperature. The resulting lysate was centrifuged at 21,000× g for 5 min in 1.5 ml tubes, and the supernatant was combined with 1.5 ml of pre-equilibrated Ni-NTA resin suspended in 4 ml of lysis buffer (20 mM Tris HCl at pH 8.0, 500 mM NaCl, and 10 mM imidazole). The mixture was incubated at 4 °C for 60 min. Subsequently, the lysate was applied to a column and washed sequentially with 2 ml of wash buffer A (20 mM Tris, 1 M NaCl), 1 ml of wash buffer B (20 mM Tris, 150 mM NaCl), and 2 ml of wash buffer C (20 mM Tris, 500 mM NaCl, and 20 mM imidazole). The protein was eluted using 1 ml of elution buffer (10 mM Tris HCl at pH 8.0, 250 mM NaCl, and 500 mM imidazole). Following elution, buffer exchange was performed by ultra-centrifugation (employing a 15 ml AMICON tube with a 50,000 molecular weight cut-off) to replace the elution buffer with 20 mM Tris at pH 8.0, 200 mM KCl, 1 mM dithiothreitol (DTT), 0.2 mM EDTA, 1% Tween 20, and 1% Igepal. The solution was then mixed with an equal volume of glycerol to yield the storage buffer, containing 10 mM Tris at pH 8.0, 100 mM KCl, 0.5 mM DTT, 0.1 mM EDTA, 0.5% Tween 20, 0.5% Igepal, and 50% glycerol. The protein concentration was determined utilizing the Pierce™ BCA Protein Assay Kit, in accordance with the manufacturer's instructions.

### Synthesis of seven-functionalized-dPTP derivatives

All seven-functionalized dPTP derivatives were synthesized and are available from Firebird Biomolecular Sciences LLC. The exact synthesis route, protocol, and all compounds' NMR and HRMS data were summarized in Supplementary information.

### Evaluate the fidelity of seven-functionalized-dPTP derivatives in 6-triphosphate PCR

ZZ sequence was chosen as a template in this evaluation model. A reaction volume of 50 μl was prepared, comprising a FAM-labeled 5' forward primer (500 nM) and a Cy5-labeled 5' reverse primer (500 nM), template ZZ (10 pM), 6-triphosphate dNTP (dATP (0.1 mM), dCTP (0.1 mM), dGTP (0.2 mM), dTTP (0.1 mM), dZTP (0.1 mM), and dPTP or functionalized dPTP (0.5 mM)), TakaRa Taq HS polymerase (1.25 U), and pH 8.0 PCR buffer. The solution was subjected to the standard 6-triphosphate PCR application condition. The resulting PCR duplexes products were analyzed and separated by urea-PAGE. The FAM-labeled ZZ chain was then subjected to the ESEGA protocol (Supplementary Fig. 12).

### Using a Z-random template assess the fidelity of seven-functionalized-dPTP derivatives in 6-triphosphate PCR

The Z-ran sequence was chosen as a template in this evaluation model. A reaction volume of 50 μl was prepared, comprising a FAM-labeled 5' forward primer (500 nM) and a Cy5-labeled 5' reverse primer (500 nM), template Z-ran (10 pM), 6-triphosphate dNTP (dATP (0.1 mM), dCTP (0.1 mM), dGTP (0.2 mM), dTTP (0.1 mM), dZTP (0.1 mM), and dPTP or functionalized dPTP (0.5 mM)), TakaRa Taq HS polymerase (1.25 U), and pH 8.0 PCR buffer. The solution was subjected to the standard 6-triphosphate PCR application condition. The resulting PCR duplexes products were analyzed and separated by urea-PAGE. The FAM-labeled ZZ chain was then subjected to the ESEGA protocol (Supplementary Fig. 13).

### Statistics and reproducibility

No statistical method was used to predetermine the sample size. The experiments were not randomized. The Investigators were not blinded to allocation during experiments and outcome assessment. The PAGE running experiment was conducted three times independently, each time yielding similar results. The translation efficiency of (ATCGZP) to (ATCG) under various PCR conditions was statistically analyzed by examining every base in the template DNA sequence, excluding the sequences of the primer regions.

### Reporting summary

Further information on research design is available in the Nature Portfolio Reporting Summary linked to this article.

## Data availability

Original gel images and numerical data generated in this study are provided in the Source data file with this paper. The raw DNA sequencing data generated in this study have been deposited in the NCBI Sequence Read Archive (SRA) database under BioProject PRJNA1104196. Source data are provided with this paper.

## Code availability

The bioinformatics analysis of the NGS data used standard software tools, such as FASTAptamer (2.0), BOWTIE2, and SAMTools, along with custom in-house scripts. The custom source code is available in Supplementary Note 1 and Zenodo[47].

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

## Acknowledgements

This work was supported by the National Institute of General Medical Sciences of the National Institutes of Health under Award Numbers R01GM141391 (S.A.B.). The content is solely the responsibility of the authors and does not necessarily represent the official views of the NIH. This work was also supported by the AEGIS LIVE Endowment Fund from the Foundation for Applied Molecular Evolution (FfAME).

## Author contributions

B.W., Z.Y., and S.B. conceived the project. B.W. designed, carried out all experiments, and wrote the manuscript. K.B. performed bioinformatics analyses. M.K. synthesized the functionally modified dPTP. R.L. and D.G. yielded the 6 M Taq polymerase variant. C.C. synthesized the DNA containing AEGIS. L.M. offers the discussion. Z.Y. and S.B. supervised the project, analyzed the data, and contributed to writing the manuscript.

## Competing interests

Many of the AEGIS components are commercially available from Firebird Biomolecular Sciences, LLC (www.firebirdbio.com), which operates under intellectual property owned by S.A.B. and the Foundation for Applied Molecular Evolution. Certain authors (K.B., M.K., and Z.Y.) are employed at Firebird. The remaining authors declare no competing interests.
