## [Peer Review File · Nature Communications]

REVIEWER COMMENTS

Reviewer #1 (Remarks to the Author):

The authors have developed a novel DNA sequencing method for their six-letter nucleotide system, utilizing the deamination of C by APOBEC, leading to a C to U mutation. In this system, the Z base is substituted with C during transliteration amplification at high pH, allowing the positions of Z in the resulting sequences to be identified as C (other original C are replaced with U by APOBEC). This approach also facilitates the quantification of Z retention in DNA amplified by PCR.

The key issue addressed is the impact of this sequencing method. If the Z-P pair proves useful for expanding the genetic alphabet as a new DNA recombinant technology, this method could become highly influential. To achieve this, it's crucial to determine the fidelity of the Z-P pair, including both unnatural to natural and natural to unnatural base mutations, in the context of six-letter DNA amplification. While the manuscript examines and describes the fidelity of the Z-P pair in PCR using various DNA polymerases, the effectiveness of this unnatural base pair remains somewhat ambiguous. One contributing factor is the use of complex DNA templates with multiple Z and P bases for fidelity evaluation. A more precise determination of Z-P pairing fidelity in PCR may be achieved using DNA templates containing a single Z or P bases, in combination with various natural base sequences.

By employing their sequencing method with simpler DNA templates containing a single Z or P, the authors should be able to more accurately ascertain the fidelity of the Z-P pair, including any sequence biases. For instance, data from Fig. 4D using KlenTaq shows fidelities of 96% and 88% (derived their source data) for Z bases at the 5' and 3' sides, respectively. This disparity likely stems from differing natural base sequences surrounding the Z bases. The authors' claim of retaining 90-95% (or 88-96% from their source data?) of Z-P pairs after 16 nominal doublings is based on the limited sequence combinations (in this case, GTGZACG, TCAZCAC, and CGCPTPGTC). However, these sequences may not represent the extremes of fidelity.

Similarly, the fidelity assessment of functionalized dPTP in six-triphosphate PCR encounters analogous issues. The authors employ a DNA template (ZZ) containing three Z bases, which could potentially lower the fidelity as compared to that utilizing DNA templates containing one Z base. Supplementary Figure 12B shows that amplification efficiencies using modified dPTPs are less than those with unmodified dPTP. If Z-P pairs mutate to natural base pairs during PCR, DNA with such mutations might amplified more efficiently than the DNA with three Z bases. Therefore, fidelity results from these ZZ-DNA experiments might be lower than those obtained from DNA with a single Z base. Furthermore, sequence biases for these modified dPTPs warrant examination.

Enhancing the characterization of the Z-P pair in replication (amplification fidelity and efficiency with different sequence combinations) by using this sequencing method could significantly broaden the impact and appeal of this sequencing method and the utility of the Z-P pair to a wider audience.

Reviewer #2 (Remarks to the Author):

This manuscript establishes a pipeline for high-throughput sequencing of DNA containing six-letters that include the four natural nucleotides and an unnatural base pair comprised of a purine analog, P, and its complementary pyrimidine analog Z. The ability to determine sequences of six-letter DNA is very powerful as the technologies that the extra nucleotides engender can be utilized to their fullest. For example, DNA with six letters has higher information density and richer chemistry due to extra functionality. As a result, libraries containing six letters have the potential to have greater success in functional selections than corresponding libraries containing only the four natural base pairs. The upstream workflow in the sequencing pipeline involves reading sequencing signatures ultimately composed of only A G C and T. This is done by carrying out pcr reactions at higher pH to deprotonate Z, which then becomes complementary to G. To remove ambiguity with C in the template, the authors compare the reads to reads obtained after all C's in the template are enzymatically deaminated to convert them to U's. The authors can also identify the location of P through inclusion of dZTP in the pcr. Moreover, the ability to sequence allows analysis of how well polymerases retain the ZP base pair in pcr reactions. This manuscript describes an enabling technology that has future impact of unleashing the power of six-letter DNA.

The manuscript is clearly presented. One issue that should be addressed has to do with dZTP incorporation. The authors clearly show that Z directs the incorporation of G at higher pH values via deprotonation. What about dZTP? Does that get incorporated opposite G during 5-triphosphate pcr and 6-triphosphate pcr? Presumably the incorporation of C opposite G is much more efficient than incorporation of Z- opposite G, but it would be helpful if the authors discuss this and how it would impact fidelity.

There are a number of typographical/grammatical errors in the text and SI.

Point-by-point list of author actions in response to Reviewer comments

Black color: Reviewer's comment

Blue color: Author's response

Reviewer #1 (Remarks to the Author):

The authors have developed a novel DNA sequencing method for their six-letter nucleotide system, utilizing the deamination of C by APOBEC, leading to a C to U mutation. In this system, the Z base is substituted with C during transliteration amplification at high pH, allowing the positions of Z in the resulting sequences to be identified as C (other original C are replaced with U by APOBEC). This approach also facilitates the quantification of Z retention in DNA amplified by PCR.

The key issue addressed is the impact of this sequencing method. If the Z-P pair proves useful for expanding the genetic alphabet as a new DNA recombinant technology, this method could become highly influential.

We deeply appreciate the reviewer's comment and his/her recognition of the impact of the expanding genetic alphabets and the sequencing technology.

To achieve this, it's crucial to determine the fidelity of the Z-P pair, including both unnatural to natural and natural to unnatural base mutations, in the context of six-letter DNA amplification.

We agree. Two "fidelities" are at issue, and it is important to distinguish them. The referee inquires here about "the fidelity of the Z-P pair", meaning the fidelity of 6-letter GACTZP PCR. It is indeed important to determine this, in both directions, unnatural to natural and natural to unnatural. We apply the new sequencing method to make this determination.

While the manuscript examines and describes the fidelity of the Z-P pair in PCR using various DNA polymerases, the effectiveness of this unnatural base pair remains somewhat ambiguous. One contributing factor is the use of complex DNA templates with multiple Z and P bases for fidelity evaluation. A more precise determination of Z-P pairing fidelity in PCR may be achieved using DNA templates containing a single Z or P bases, in combination with various natural base sequences.

We agree. The template contains both isolated Z:P pairs, as well as consecutive Z:P pairs. Experience has shown that local sequence context extends over a few pairs. Thus, an isolated Z:P pair more than 5 or so nucleotides distant from a second Z:P pair is effectively the same as a single Z:P pair in a template.

By employing their sequencing method with simpler DNA templates containing a single Z or P, the authors should be able to more accurately ascertain the fidelity of the Z-P pair, including any sequence biases.

We agree. To expand the number of examples, we synthesized three sequences where a single C, Z or P was placed in the middle of six random nucleotides (C-Ran, Z-Ran and P-Ran, Table 1). The ESEGA workflow was applied. Low sequence bias was seen in the deamination results (Fig. 3H, top), consistent with the literature. Further, no overall obvious sequence context bias was observed in Z and P transliteration (Fig. 3H, middle and bottom). (the details results reply to Source data fold: sequence bias analysis).

C-Ran	TAAGATGAGAGTTGAGGAGAGTTATNNNCNNNGTATAGTAGTGTAAAGTAGATAGTGGA
Z-Ran	TAAGATGAGAGTTGAGGAGAGTTATNNNZNNNGTATAGTAGTGTAAAGTAGATAGTGGA
P-Ran	TAAGATGAGAGTTGAGGAGAGTTATNNNPNNNGTATAGTAGTGTAAAGTAGATAGTGGA

For instance, data from Fig. 4D using KlenTaq shows fidelities of 96% and 88% (derived their source data) for Z bases at the 5' and 3' sides, respectively. This disparity likely stems from differing natural base sequences surrounding the Z bases. The authors' claim of retaining 90-95% (or 88-96% from their source data?) of Z-P pairs after 16 nominal doublings is based on the limited sequence combinations (in this case, GTGZACG, TCAZCAC, and CGCPTPGTC). However, these sequences may not represent the extremes of fidelity.

We agree. Here again, we are speaking of the fidelity of PCR, not the accuracy of the sequencing methods. While we cannot example all possible sequences, we expect that those presented approximate a reasonable range of fidelity extremes. To expand that range, we added an experiment using Z-Ran templates (-NNNZNNN-) followed by a 6-triphosphate PCR. To evaluate the fidelity of Z/P retention post the 6-letter PCR, we employed the ESEGA sequencing method.

The obtained results for the sequence -NNNZNNN- under various conditions are as follows:

1. With dATP, dTTP, dCTP, dGTP, dZTP, and d^PTP conditions, the fidelity of the 6-triphosphate PCR was found to be 94.03%.
2. When dATP, dTTP, dCTP, dGTP, dZTP, and d^{eth}P^TP were used, the PCR fidelity decreased to 83.46%.
3. In the presence of dATP, dTTP, dCTP, dGTP, dZTP, and d^{ph}P^TP, the fidelity further reduced to 81.19%.

Following PCR, the products were deaminated, causing C to transform into U within the sequence -NNNZNNN-. A detailed analysis of sequencing biases stemming from this alteration was not available. Detailed experimental conditions and procedures are included in the Supporting Information.

Supplementary Figure 13 | Z-rand template assesses the fidelity of different modified dPTPs in 6-triphosphate PCR.

Similarly, the fidelity assessment of functionalized dPTP in six-triphosphate PCR encounters analogous issues. The authors employ a DNA template (ZZ) containing three Z bases, which could potentially lower the fidelity as compared to that utilizing DNA templates containing one Z base. Supplementary Figure 12B shows that amplification efficiencies using modified dPTPs are less than those with unmodified dPTP. If Z-P pairs mutate to natural base pairs during PCR, DNA with such mutations might amplified more efficiently than the DNA with three Z bases. Therefore, fidelity results from these ZZ-DNA experiments might be lower than those obtained from DNA with a single Z base. Furthermore, sequence biases for these modified dPTPs warrant examination.

Again, we are using the new sequencing method to determine the fidelity of GACTZP PCR amplification, sampling many contexts including those that may not be replicated with high fidelity. The reviewer highlights a crucial aspect of the 6-letter PCR process. (Reviewer point out that if Z-P pairs mutate to natural base pairs during PCR, DNA with such mutations might be amplified more efficiently than DNA containing three Z bases). This is key issue in 6-letter aptamer selection is analyzed using the sequencing method reported here.

In this context, the choice of an AEGIS template containing double ZZ was intentional. Our goal was to assess the fidelity of the 6-letter PCR in the context of relatively challenging sequences. The results indicate that when using modified dPTP, the fidelity at the double ZZ positions decreased. This finding is significant for our project and will guide future improvements in our approach.

We also add an experiment that using –NNNZNNN- to assess the fidelity of modified dPTP in 6-letter PCR, as mentioned in the last response.

Enhancing the characterization of the Z-P pair in replication (amplification fidelity and efficiency with different sequence combinations) by using this sequencing method could significantly broaden the impact and appeal of this sequencing method and the utility of the Z-P pair to a wider audience.

We agree. Thank you so much.

Reviewer #2 (Remarks to the Author):

This manuscript establishes a pipeline for high-throughput sequencing of DNA containing six-letters that include the four natural nucleotides and an unnatural base pair comprised of a purine analog, P, and its complementary pyrimidine analog Z. The ability to determine sequences of six-letter DNA is very powerful as the technologies that the extra nucleotides engender can be utilized to their fullest. For example, DNA with six letters has higher information density and richer chemistry due to extra functionality. As a result, libraries containing six letters have the potential to have greater success in functional selections than corresponding libraries containing only the four natural base pairs. The upstream workflow in the sequencing pipeline involves reading sequencing signatures ultimately composed of only A G C and T. This is done by carrying out PCR reactions at higher pH to deprotonate Z, which then becomes complementary to G. To remove ambiguity with C in the template, the authors compare the reads to reads obtained after all C's in the template are enzymatically deaminated to convert them to U's. The authors can also identify the location of P through inclusion of dZTP in the PCR. Moreover, the ability to sequence allows analysis of how well polymerases retain the ZP base pair in PCR reactions. This manuscript describes an enabling technology that has future impact of unleashing the power of six-letter DNA.

We greatly appreciate the reviewer's comment.

The manuscript is clearly presented. One issue that should be addressed has to do with dZTP incorporation. The authors clearly show that Z directs the incorporation of G at higher pH values via deprotonation. What about dZTP? Does that get incorporated opposite G during 5-triphosphate PCR and 6-triphosphate PCR? Presumably the incorporation of C opposite G is much more efficient than incorporation of Z opposite G, but it would be helpful if the authors discuss this and how it would impact fidelity.

The reviewer raises an important question regarding the behavior of dZTP in the PCR process. In the 5-triphosphate PCR, the incorporation of C opposite G is more efficient than incorporation of dZTP opposite G, at the same time there are still few dZTP may pair opposite G, but because dPTP is absent, all Z bases will pair opposite G in subsequent PCR rounds. Therefore, in this context, dZTP does not disrupt the fidelity of C:G pairings. Our result also shows the C:G retention close to 99.8% in 5-triphosphate PCR.

However, in the 6-triphosphate PCR, the situation is as the reviewer suggests: few dZTP can pair with G, potentially leading to Z/P pairs replacing C:G pairs. This occurrence depends on the specific polymerase used. In Figure 4D, we clearly observe that when using KOD- polymerase, P partially replaces G. This replacement is much much weaker when using Taq polymerase. This indicates that the choice of polymerase significantly influences the outcome of the PCR process in the context of incorporating synthetic bases.

There are a number of typographical/grammatical errors in the text and SI.
We fixed some.

** ---

REVIEWERS' COMMENTS

Reviewer #1 (Remarks to the Author):

The manuscript has been updated in response to comments from the reviewers.

For an enhanced Introduction section of this manuscript, it is essential to note that in the field of unnatural bases, the group led by Hirao has not only contributed significantly by developing a high-throughput sequencing approach but has also effectively utilized this technique for generating aptamers using their Ds-Px pair. Predominantly focused on applications involving a five-letter genetic code, their recent publication in *Nucleic Acids Research* in 2021 showcases the extension of their methodology to include a six-letter aptamer. This aptamer's sequence was determined by integrating the transliteration method with modified Sanger sequencing techniques. While the authors do cite several of their publications, it is crucial for the Introduction section to accurately and comprehensively present their significant contributions to the field.

Reviewer #2 (Remarks to the Author):

The authors have satisfactorily addressed the point about G possibly directing the incorporation opposite deprotonated dZTP in the response. They may want to consider adding this discussion to the manuscript or Supporting Information.